# Development of Novel Paclitaxel-Loaded ZIF-8 Metal-Organic Framework Nanoparticles Modified with Peptide Dimers and an Evaluation of Its Inhibitory Effect against Prostate Cancer Cells

**DOI:** 10.3390/pharmaceutics15071874

**Published:** 2023-07-03

**Authors:** Heming Zhao, Liming Gong, Hao Wu, Chao Liu, Yanhong Liu, Congcong Xiao, Chenfei Liu, Liqing Chen, Mingji Jin, Zhonggao Gao, Youyan Guan, Wei Huang

**Affiliations:** 1State Key Laboratory of Bioactive Substance and Function of Natural Medicines, Department of Pharmaceutics, Institute of Materia Medica, Chinese Academy of Medical Sciences and Peking Union Medical College, Beijing 100050, China; zhaoheming@imm.ac.cn (H.Z.); dawngong@163.com (L.G.); chaoliu@imm.ac.cn (C.L.); liuyanhong@imm.ac.cn (Y.L.); xiaocongcong@imm.ac.cn (C.X.); liuchenfei@imm.ac.cn (C.L.); chenliqing@imm.ac.cn (L.C.); jinmingji@imm.ac.cn (M.J.); 2Department of Pharmacy, Yanbian University, Yanji 133000, China; wuhao931230@163.com; 3Department of Urology, National Cancer Center/National Clinical Research Center for Cancer/Cancer Hospital, Chinese Academy of Medical Sciences and Peking Union Medical College, Beijing 100021, China

**Keywords:** metal-organic framework, paclitaxel, peptide dimer, prostate cancer

## Abstract

Prostate cancer (PC) is one of the common malignant tumors of the male genitourinary system. Here, we constructed PTX@ZIF-8, which is a metal-organic-framework-encapsulated drug delivery nanoparticle with paclitaxel (PTX) as a model drug, and further modified the synthesized peptide dimer (Di-PEG_2000_-COOH) onto the surface of PTX@ZIF-8 to prepare a nanotargeted drug delivery system (Di-PEG@PTX@ZIF-8) for the treatment of prostate cancer. This study investigated the morphology, particle size distribution, zeta potential, drug loading, encapsulation rate, stability, in vitro release behavior, and cytotoxicity of this targeted drug delivery system, and explored the uptake of Di-PEG@PTX@ZIF-8 by human prostate cancer Lncap cells at the in vitro cellular level, as well as the proliferation inhibition and promotion of apoptosis of Lncap cells by the composite nanoparticles. The results suggest that Di-PEG@PTX@ZIF-8, as a zeolitic imidazolate frameworks-8-loaded paclitaxel nanoparticle, has promising potential for the treatment of prostate cancer, which may provide a novel strategy for the delivery system targeting prostate cancer.

## 1. Introduction

Prostate cancer (PC) is one of the common malignant tumors of the male genitourinary system, which has become the second most prevalent malignant tumor around the world [1]. Antiandrogen therapy plays an integral role in the treatment of prostate cancer. Orchiectomy is a traditional antiandrogen therapy with low cost, relatively few side effects, and the capacity to rapidly reduce and maintain low levels of testosterone [2]. In the early stage, prostate cancer can be treated surgically, but most prostate cancers are already in the progressive stage at the time of diagnosis, due to the lack of specific symptoms in the early stages of prostate cancer [3], making complete surgical removal of the tumor almost impossible. Also, most patients struggle to accept such a form of treatment. Some hormonal drugs including estrogens, gonadotropin-releasing hormone analogs, gonadotropin-releasing hormone antagonists, androgen biosynthesis inhibitors, and androgen receptor blockers, which can also achieve lower testosterone, but they have significant side effects and, as the disease progresses, almost all patients will no longer be sensitive to androgens and develop desmoplastic-resistant prostate cancer [4]. Therefore, the treatment of tumors with anticancer drugs has an irreplaceable place in clinical practice. Chemotherapy is one of the most extensively available methods to combat advanced prostate cancer, but its therapeutic efficacy is often inadequate due to poor specificity and associated toxicity [5]. Known as a broad-spectrum antitumor drug, paclitaxel (PTX) has become the mainstay of chemotherapy for clinical practice, as first- or second-line treatment for several types of cancer [6]. Paclitaxel blocks cell mitosis mainly by inducing microtubule protein polymerization, thus inhibiting proliferation and eventually promoting the apoptosis of tumor cells [7]. However, paclitaxel shows low aqueous solubility [8], poor bioavailability [9], and an inability to effectively differentiate between normal and tumor tissues, thereby suffering from strong adverse effects. To address the challenges of the clinical application of PTX, a large number of studies have been conducted to adopt nanocarriers for targeting delivery to tumor sites, improving pharmacokinetics, and reducing systemic toxicity.

Metal-organic framework materials (MOFs), a type of nanocarrier emerging in recent years, are porous materials formed by coordination between organic skeletons and metal ions [10,11,12], with porosity, large specific surface area, and high adsorption properties, thus adopted for the delivery of insoluble drugs. Zeolitic imidazolate frameworks-8 (ZIF-8) is a MOF material with a tetrahedral space structure formed by the N atom in dimethylimidazole linked to the zinc ion through the coordination bond [13,14]. Compared with traditional drug delivery carriers such as liposomes, nanomicelles, and polymeric nanoparticles, ZIF-8 has the advantages of high stability and high encapsulation rate for chemotherapeutic drugs. Furthermore, ZIF-8 is nontoxic and has good biocompatibility and biodegradability [15]. ZIF-8 has the pH-responsive biodegradation property [16], so the adoption of ZIF-8 as a drug delivery carrier ensures stability in blood circulation to avoid toxicity caused by premature drug release, and once entering the tumor cells, it will disintegrate in an acidic endosomal/lysosomal environment for controlled release of chemotherapeutic drugs. Therefore, ZIF-8 is selected as a carrier material for chemotherapeutic drugs to construct a tumor precision-targeted drug delivery system, which can achieve the aim of reducing toxicity and increasing efficiency [17].

With the rapid development of proteomics, phage display, and peptide solid-phase synthesis in recent years, more and more novel peptides have been discovered or rationally designed, which greatly facilitate the development of targeted drug delivery systems based on peptides. Compared with the monomer of the peptide, the polypeptide significantly enhances the affinity and specificity with the receptor, with higher targeting efficiency to effectively avoid the off-target effect [18]. Peptide homodimers or multimers are formed by coupling identical types of peptides through suitable linkers [19]. Due to the bivalent effect of homodimers, peptide dimers can obtain a higher receptor affinity for homodimers, better tumor uptake, and longer tumor retention time [20]. Di-peptide, the dimerization of peptides (WQPDTAHHWATL) through the use of a lysine residue at the COOH terminus, was found to exhibit stronger specific affinity for prostate-specific membrane antigen (PSMA), which is highly expressed on the surface of prostate cancer cells [21]. Also, Zhang et al. [22] in our lab have also demonstrated the specificity of this peptide for Lncap cells in a previous study. Accordingly, Di-peptide was served as the targeting peptide for the construction of nanodelivery systems targeting prostate cancer with a high affinity for PSMA.

In this study, PTX@ZIF-8 nanoparticles were synthesized by the one-pot method, and then Di-PEG-COOH was modified on the surface of PTX@ZIF-8 by positive and negative charge interactions to obtain the targeted metal-organic framework drug delivery system Di-PEG@PTX@ZIF-8. This study investigated the physicochemical properties of nanoparticles in terms of morphology, particle size distribution, zeta potential, drug loading, encapsulation rate, stability, and in vitro release behavior. In addition, the cytotoxicity and uptake of Di-PEG@PTX@ZIF-8 nanoparticles on human prostate cancer Lncap cells were studied at the in vitro cellular level, as well as the promotion of apoptosis and proliferation inhibition of Lncap cells via the composite nanoparticles. It was concluded that Di-PEG@PTX@ZIF-8 nanoparticles could serve as a drug delivery platform with great potential to treat prostate cancer.

## 2. Materials and Methods

### 2.1. Materials

PTX was purchased from Shanghai Aladdin Biochemical Technology Co., Ltd. (Shanghai, China). Maleimide-[(polyethyleneglycol)_2000_]-carboxylic acid (Mal-PEG_2000_-COOH) and Methoxy-[(polyethyleneglycol)_2000_]- carboxylic acid (mPEG_2000_-COOH) were brought from AVT Shanghai Pharmaceutical Tech Co., Ltd. (Shanghai, China). The polypeptide dimer (CWQPDTAHHWATL)_2_K was custom-synthesized by Tanshui-Tech (Guangzhou, China). 2-Methylimidazole was purchased from Beijing Huabozhan Bioanalytical Technology Co., Ltd. (Beijing, China). Zinc nitrate hexahydrate was bought from Lanzhou yellow river institute of Zinc and Magnesium Nanomaterials (Lanzhou, China). Coumarin 6 (C6) was purchased from J&K Scientific Co., Ltd. (Beijing, China). Trypsin, RPMI 1640 medium and phosphate buffer saline (PBS) were purchased from Thermo Fisher Scientific Co. (Beijing, China). Cell counting kit-8 (CCK-8) was obtained from Dojindo Laboratories (Kumamoto, Japan). All reagents were analytical grade and used without further purification.

### 2.2. Cell Lines and Cell Culture

Human prostate cancer cells (Lncap cells) were purchased from the Cell Culture Center of Institute of Basic Medical Sciences in Chinese Academy of Medical Sciences (CAMS, Beijing, China). The Lncap cell lines were cultured in Roswell Park Memorial Institute-1640 (RPMI-1640) medium supplemented with 10% (*v*/*v*) fetal bovine serum (FBS) and 1% (*v*/*v*) penicillin–streptomycin (Pen-Strep) in a humidified 5% CO_2_ atmosphere at 37 °C.

### 2.3. Synthesis and Characterization of Di-PEG-COOH

Di-PEG-COOH was synthesized by conjugating the cysteine residues of the peptide dimer to Mal-PEG-COOH as reported in the literature [23]. Briefly, 45 mg of Di-peptide and 40 mg of Mal-PEG-COOH (molar ratio 1:2) were weighed in a 50 mL round bottom flask, and then 10 mL of pH 8.0 HEPEs buffer (0.1 M) was added and filled with nitrogen for protection. The whole reaction system was stirred at room temperature for 20 h. The reaction mixture was dialyzed in distilled water with a molecular weight cut-off (MWCO) of 1000 for 24 h to remove the excess peptides. The final product Di-PEG-COOH was obtained by cryodesiccation. The structure of Di-PEG-COOH was confirmed by ^1^H nuclear magnetic resonance (NMR) spectroscopy (400 MHz, Varian Medical Systems, Inc., Palo Alto, CA, USA). Matrix-assisted laser desorption ionization time of flight-mass spectrometry (MALDI-TOF-MS) (4800 Plus, Applied Biosystems Inc., Waltham, MA, USA) was used for further analysis of Di-peptide, Mal-PEG-COOH, and Di-PEG-COOH, respectively.

### 2.4. Preparation of PTX@ZIF-8 and Di-PEG@PTX@ZIF-8

The PTX@ZIF-8 nanoparticles were prepared by the “one-pot method”. Briefly, 10 mg of paclitaxel and 80 mg of zinc nitrate hexahydrate were dissolved in 4 mL dimethyl sulfoxide in a 10 mL test tube under ultrasound and stirred for 30 min. Then, 4 mL aqueous solution containing 800 mg of 2-methylimidazole was added to it and reacted for 30 min at room temperature. The reaction solution was centrifuged at 14,000 rpm/min for 14 min and then the reaction solution was washed three times with methanol to obtain PTX@ZIF-8. Di-PEG@PTX@ZIF-8 was prepared by dissolving 4 mg of Di-PEG-COOH and 20 mg of PTX@ZIF-8 (mass ratio 1:5) in 5 mL of deionized water and stirring at 550 rpm/min for 16 h.

### 2.5. Characterization of PTX@ZIF-8 and Di-PEG@PTX@ZIF-8

The particle size distribution and zeta potential of PTX@ZIF-8 and Di-PEG@PTX@ZIF-8 were determined using dynamic light scattering (DLS) and electrophoretic light scattering (Zetasizer Nano ZS90; Malvern Instruments Ltd., Malvern, UK). The morphologies of PTX@ZIF-8 and Di-PEG@PTX@ZIF-8 were examined using transmission electron microscopy (JEM-1400PLUS, JEOL Ltd., Tokyo, Japan).

The drug loading capacity (DL%) and encapsulation efficiency (EE%) were measured by high-performance liquid chromatography (HPLC, Agilent 1260 infinity; Agilent Technologies, Santa Clara, CA, USA). Briefly, PTX was dissolved in methanol and injected into an Agilent HPLC-C18 column (5 μm, 250 × 4.6 mm), with a mobile phase of water:methanol:acetonitrile = 41:23:36, and detected at a wavelength of 227 nm with a column temperature of 35 °C, a flow rate of 1 mL/min, and a sample volume of 10 μL. The Di-PEG@PTX@ZIF-8 NPs were centrifuged at 6000 rpm for 8 min, and the supernatant was collected and diluted to 10 mL with methanol for the determination of PTX content (W_1_). The same volume of uncentrifuged Di-PEG@PTX@ZIF-8 NPs solution was added with the appropriate amount of pH 2.0 hydrochloric acid solution to completely destroy the nanoparticle structure, and similarly diluted to 10 mL with methanol for the determination of PTX content (W_2_). The same volume of Di-PEG@PTX@ZIF-8 NPs solution was freeze-dried to obtain its weight (W_3_). The encapsulation efficiency (EE) and drug loading (DL) of Di-PEG@PTX@ZIF-8 NPs were calculated according to the following formulas:$$\mathrm{EE}\;\left(\%\right)=\frac{W_{2}-W_{1}}{W_{2}}\times 100\%$$
$$\mathrm{DL}\;\left(\%\right)=\frac{W_{2}-W_{1}}{W_{3}}\times 100\%$$

### 2.6. Stability Examination of Di-PEG@PTX@ZIF-8

The stability of Di-PEG@PTX@ZIF-8 NPs was analyzed using a Zetasizer Nano ZS90 instrument. The particle sizes and polydispersity index (PDI) of the initial Di-PEG@PTX@ZIF-8 NPs were measured. Subsequently, Di-PEG@PTX@ZIF-8 was incubated in 10 mmol/L PBS (pH 7.4) and stored at 4 °C. The particle size of the sample solution was measured and recorded on days 1, 2, 3, 4, 5, 6, and 7, respectively.

### 2.7. In Vitro Release Assay

The in vitro drug release behavior of Di-PEG@PTX@ZIF-8 was evaluated using the dialysis diffusion technique. A total of 1 mL of Di-PEG@PTX@ZIF-8 NPs solution, containing PTX at a concentration of 500 μg/mL, was transferred to the dialysis bag (MWCO 1000D), which was placed into the release medium of 20 mL PBS solution (pH 7.4, containing 0.5% Tween-80) and 20 mL PBS solution (pH 5.5, containing 0.5% Tween-80), respectively, and shaken at 37 °C with a constant temperature and a gyration speed of 100 rpm. Then, 0.5 mL of release medium was removed at 0.5 h, 1 h, 2 h, 4 h, 8 h, 12 h, 16 h, 24 h, 36 h, and 48 h, while 0.5 mL of the corresponding release medium was replenished. The removed medium was centrifuged at 12,000 rpm for 10 min, and the supernatant was taken to determine the drug concentration via HPLC and calculate the accumulative drug release.

### 2.8. In Vitro Cellular Uptake Studies

Coumarin-6 (C6) was employed to replace PTX to observe the cellular uptake of Coumarin-6-labeled nanoparticles with or without Di-PEG (mPEG@Cou-6@ZIF-8 and Di-PEG@Cou-6@ZIF-8) in vitro. Lncap cells were obtained at a density of 2 × 10^5^/well in a 12-well plate and incubated at 37 °C in a 5% CO_2_ incubator for 24 h. Then, the old culture medium was discarded and incubated with serum-free RPMI-1640 medium diluted with free coumarin 6 (Cou-6) solution, mPEG@Cou-6@ZIF-8, and Di-PEG@Cou-6@ZIF-8, all at a concentration of 2 μg/mL of Cou-6. All groups were placed in the incubator for 0.5 h, 1 h, 1.5 h, and 2 h. The old medium containing the drug was aspirated and cold PBS (pH 7.4) was added to terminate the cellular uptake process. The cells were washed three times with PBS (pH 7.4) and then fixed with 4% paraformaldehyde before staining the nuclei with DAPI. After being treated with an antifluorescence quenching agent, confocal laser scanning microscopy (CLSM) was performed to observe the penetration of drugs and nanoparticles into cells at different culture times.

In addition, flow cytometry (FCM, Becton Dickinson, Franklin Lake, NJ, USA) was used to examine the cellular uptake efficiency quantitatively. Lncap cells were cultured in 6-well plates at a density of 4 × 10^5^/well for 24 h at 37 °C and 5% CO_2_. Then, the old medium was aspirated, and after washing once with serum-free RPMI-1640 medium, 2 mL of serum-free RPMI-1640 medium solutions containing free Cou-6 solution, mPEG@Cou-6@ZIF-8, and Di-PEG@Cou-6@ZIF-8 were added to each well, all at a concentration of 1 μg/mL of C6. All groups were placed in the incubator for 0.5 h, 1 h, 1.5 h, and 2 h. The old medium was aspirated and washed 3 times with cold PBS. After digestion with 0.25% trypsin, the cells were centrifuged and resuspended with 0.5 mL PBS. The cell uptake was detected using flow cytometry and screened for the appropriate time to perform targeting experiments.

### 2.9. Cell Viability Assay

CCK-8 assays were performed to evaluate the cytotoxicity of blank nanoparticles on Lncap cells. In brief, Lncap cells in the logarithmic growth phase were inoculated in 96-well plates at a density of 12,000/well, with a volume of 200 μL per well, and incubated at 37 °C and 5% CO_2_ for 24 h. Then, 200 μL of culture solution containing only Di-PEG@ZIF-8 corresponding to PTX concentrations of 10, 5, 1, 0.5, 0.1, 0.05, 0.01, 0.001 μg/mL were added, respectively, and continued to incubate for 24 h and 48 h. After the incubation, the culture solution was aspirated and 100 μL of serum-free culture solution containing 10% CCK-8 was added to each well and incubated for 4 h. The wells without Di-PEG@ZIF-8 were designated as control wells, and 6 replicate wells were set up for each well. The OD value at 450 nm was measured using a Synergy H1 Microplate Reader (BioTek, Dallas, TX, USA). Cell viability was calculated as follows:$$\mathrm{Cell}\;\mathrm{viability}\;\left(\%\right)=\frac{{\mathrm{OD}}_{\mathrm{test}}-{\mathrm{OD}}_{\mathrm{blank}}}{{\mathrm{OD}}_{\mathrm{control}}-{\mathrm{OD}}_{\mathrm{blank}}}\times 100\%$$

CCK-8 assay was employed to examine the inhibition of cell proliferation by free PTX, mPEG@PTX@ZIF-8, and Di-PEG@PTX@ZIF-8 on Lncap cells. Lncap cells were inoculated in 96-well plates at a density of 12,000 cells/200 μL/well and incubated at 37 °C in a 5% CO_2_ incubator for 24 h. Then, the old culture solution was aspirated and three groups of PTX, mPEG@PTX@ZIF-8, and Di-PEG@PTX@ZIF-8 were set up to dose at concentrations of 10, 5, 1, 0.5, 0.1, 0.05, 0.01, and 0.001 μg/mL, respectively. Using the wells without drug action as control wells, six replicate wells were set up for each well and incubated for 24 h and 48 h, respectively. After the incubation, the old culture solution was aspirated and 100 μL of serum-free medium containing 10% CCK-8 was added under dark conditions and incubated at 37 °C and 5% CO_2_ for 4 h. The absorption values of each well at 450 nm were measured with a Synergy H1 Microplate Reader (BioTek, Dallas, TX, USA). The cell survival rate was calculated according to the above formula.

### 2.10. Wound-Healing Assay

Wound-healing assays were performed to evaluate the effect of different preparations on the migration and motility of Lncap cells. After incubation of Lncap cells in 6-well plates for 24 h, a wound area was traced in the middle of the wells and washed with PBS to remove floating cells. Then, free PTX, mPEG@PTX@ZIF-8, and Di-PEG@PTX@ZIF-8 NPs were added at a PTX concentration of 2 µg/mL, and fresh medium was used as the control group. The status of the wound area was observed after 0, 12, 24, and 48 h with an inverted light microscope (Olympus, Hamburg, Germany).

### 2.11. Cell Apoptosis Assay

Hoechst staining assay was employed to analyze the cell apoptosis of free PTX, mPEG@PTX@ZIF-8, and Di-PEG@PTX@ZIF-8 on Lncap cells qualitatively. Lncap cells were inoculated in 12-well plates at a density of 2 × 10^5^/well and incubated at 37 °C with 5% CO_2_ for 24 h. After aspirating the old medium, 1 mL of PTX, mPEG@PTX@ZIF-8, and Di-PEG@PTX@ZIF-8 containing 2 µg/mL PTX were added, respectively, and a blank control group was set up, continuing to incubate for 48 h. Then, 4% paraformaldehyde was added to fix for 15 min, and the nuclei were stained with Hoechst 33258 (5 μg/mL). The morphological changes of the nuclei were observed under an IX51 inverted fluorescence microscope (Olympus Corporation, Tokyo, Japan).

The apoptosis of Lncap cells was quantitatively analyzed via Annexin V-FITC/PI double staining assay. Lncap cells were inoculated in 6-well plates at a density of 4 × 10^5^ cells/mL/well and incubated at 37 °C and 5% CO_2_ for 24 h. Then, the old culture solution was aspirated and 2 mL of PTX, mPEG@PTX@ZIF-8, and Di-PEG@PTX@ZIF-8 containing 2 µg/mL PTX were added, and a blank control group was set up, continuing to incubate for 48 h. The cells were digested with 0.25% EDTA-free trypsin, resuspended with 0.5 mL binding buffer, and then 5 μL Annexin V-FITC and 10 μL propidium iodide (PI) staining solution were added. After 10 min, the percentage of apoptosis cells was analyzed immediately with a FACSCalibur flow cytometer (Becton Dickinson, Franklin Lake, NJ, USA).

### 2.12. Statistical Analysis

All data were reported as mean ± standard deviation (SD). Statistical analysis of the data was performed using GraphPad Prism software 8.0 with two-tailed Student’s *t*-test or one-way analysis of variance (ANOVA). Statistical significance was indicated as (n. s.) *p* > 0.05, (*) *p* < 0.05, (**) *p* < 0.01, (***) *p* < 0.001, and (****) *p* < 0.0001.

## 3. Results

### 3.1. Synthesis of Di-PEG-COOH

The Di-PEG-COOH was prepared as shown in Figure 1A, in which Di-peptide was conjugated to the maleimide group of Mal-PEG-COOH via the sulfhydryl group of the terminal cysteine. The products were characterized by ^1^H NMR and MALDI-TOF-MS to confirm the structure. Figure 1B shows the ^1^H NMR spectra of Mal-PEG-COOH and Di-PEG-COOH in D_2_O, respectively. The characteristic absorption peak of the maleimide group in Mal-PEG-COOH appeared at 6.7 ppm, which disappeared in the ^1^H NMR spectrum of Di-PEG-COOH, indicating its successful synthesis. In addition, as shown in Figure 1C, the major peaks of Mal-PEG-COOH, Di-peptide, and Di-PEG-COOH in MALDI-TOF-MS were 2129.12, 3241.55, and 7351.24, respectively, in which the molecular weight of the synthesized Di-PEG-COOH was basically consistent with the theoretical value of 7331.55. All the above results indicated the successful synthesis of Di-PEG-COOH.

### 3.2. Characterization of PTX@ZIF-8 and Di-PEG@PTX@ZIF-8

In view of the unique physiological environment of the tumor tissues, the suitable particle size and potential serve an instrumental role in the aggregation of nanoparticles at the tumor site. DLS measurements demonstrated that PTX@ZIF-8 had an average particle size of 107.4 ± 0.26 nm (Figure 2A), PDI of 0.024 ± 0.01, and zeta potential of 31.5 ± 2.1 mV (Figure 2B), while Di-PEG@PTX@ZIF-8 showed an average size of 126.1 ± 0.80 nm (Figure 2D), PDI of 0.144 ± 0.01, and zeta potential of −12.3 ± 1.22 mV (Figure 2E). The drug-loaded nanoparticles PTX@ZIF-8 were conjugated with Di-PEG-COOH by the “electrostatic self-assembly method” to obtain Di-PEG@PTX@ZIF-8 NPs, which resulted in the increase of the particle size and the change of the surface charge of the nanoparticles from positive to negative charge. Studies have indicated that nanoparticles with a slight negative charge on their surface contribute to avoiding uptake by non-target tissues and are easy to accumulate in tumors. The morphology of PTX@ZIF-8 (Figure 2C) was observed under transmission electron microscopy as showing angular dodecahedra, while the surface of Di-PEG@PTX@ZIF-8 (Figure 2F) had blunted angles and no longer remained clear dodecahedra, but seemed to be covered with a film, indicating that Di-PEG-COOH had covered the surface of PTX@ZIF-8 through electrostatic interactions to form the Di-PEG@PTX@ZIF-8 NPs. In addition, the encapsulation rate of Di-PEG@PTX@ZIF-8 NPs was 55.75% ± 0.56% and the drug loading capacity was 12.8% ± 0.31%. As shown in Figure 2G, there was no significant change in the particle size and PDI of both PTX@ZIF-8 and Di-PEG@PTX@ZIF-8 at 4 °C for one week, indicating good stability of nanoparticles in PBS (pH 7.4). Compared with PTX@ZIF-8, Di-PEG@PTX@ZIF-8 tends to agglomerate at high concentrations and becomes less stable after 7 days, which may be due to the fact that the nanoparticle surface is not fully encapsulated and the bare surface is positively charged, which inter-adsorbs with nearby nanoparticles that have been encapsulated by Di-PEG-COOH and have a negatively charged surface. However, at low concentrations, the stability was relatively favorable and did not agglomerate. Therefore, during blood circulation, low-concentration drugs do not appear to aggregate, while after targeting and accumulating in tumor tissues, high-concentration drugs achieve agglomeration and have stronger retention. Furthermore, Di-PEG@PTX@ZIF-8 NPs also exhibited good stability in three different media: deionized water, RPMI-1640 medium, and 5% glucose (Figure 2H), which further illustrated the outstanding stability of the nanoparticles.

### 3.3. In Vitro Drug Release

In order to investigate the in vitro release behavior of PTX from nanoparticles, Di-PEG@PTX@ZIF-8 NPs were placed in PBS solution (containing 0.5% Tween-80) at pH 7.4 and pH 5.5, respectively, to simulate the normal physiological environment and the acidic environment at tumor tissues, and explored the cumulative release at different time points within 48 h at 37 °C. As shown in Figure 2I, Di-PEG@PTX@ZIF-8 showed rapid degradation of nanoparticles and nearly complete release of the drug at pH 5.5, reaching 60% at 16 h, while little release was achieved at pH 7.4. The release profile of the drug was found to correspond to the first-order kinetic model after fitting. The fitted equation for the release profile at pH 7.4 was $y=20.96\times \left(1-e^{-0.19x}\right)$ and R^2^ was 0.91. The fitted equation was $y=47.31\times \left(1-e^{-0.55x}\right)$ and R^2^ was 0.96 for the release curve at pH 5.5. Therefore, Di-PEG@PTX@ZIF-8 NPs manage to release less of the drug in blood circulation, reducing the side effects and toxicity of drugs to other tissues and organs, while in the acidic environment of tumor tissues, Di-PEG@PTX@ZIF-8 NPs can rapidly cleave and release the drug to reach the effective concentration and exert the tumor-killing effect.

### 3.4. Cellular Uptake Study

In this study, coumarin-6 was employed to replace PTX in the preparation of mPEG@Cou-6@ZIF-8 and Di-PEG@Cou-6@ZIF-8 to investigate their uptake by Lncap cells. The results of cellular uptake were observed with CLSM and analyzed quantitatively using flow cytometry. Figure 3A demonstrated the cellular uptake of Di-PEG@Cou-6@ZIF-8 NPs at different times. Blue and green represent the fluorescent signals of the nucleus and C6, respectively. The fluorescence signal of C6 could be observed after 0.5 h incubation of Di-PEG@Cou-6@ZIF-8 NPs and appeared to be time-dependent, with a significant increase in fluorescence intensity with increasing time. The quantitative analysis of cell uptake results from the flow cytometry assay also revealed similar results to that obtained with CLSM. As observed from Figure 3B,C, the cellular uptake of Lncap cells incubated with Di-PEG@Cou-6@ZIF-8 NPs for 2 h reached a high level and was largely saturated.

Given the above experimental results, 2 h was chosen as the incubation time to examine the uptake of different formulations by Lncap cells. To exclude the interference of PEG on the experimental results, three groups were set up for cellular uptake assays: free C6 group, mPEG@Cou-6@ZIF-8 NPs group, and Di-PEG@Cou-6@ZIF-8 NPs group. As can be seen from Figure 4A, the green fluorescence signals were barely observed from the free C6 group, and there were only little green fluorescence signals from the mPEG@Cou-6@ZIF-8 NPs group, which means that free C6 were barely taken up by Lncap cells, and only a small amount of C6 encapsulated by mPEG@ZIF-8 was taken up. In contrast, the green fluorescent signal could be clearly observed in the Di-PEG@Cou-6@ZIF-8 group, indicating that the Di-PEG@Cou-6@ZIF-8 NPs were taken up by Lncap cells with high efficiency. The flow cytometry assays demonstrated more intuitive results for the quantitative analysis of cellular uptake. Di-PEG@Cou-6@ZIF-8 showed the highest cellular uptake level on Lncap cells, with about 156-fold higher than free C6 and nearly 1.8-fold higher than mPEG@Cou-6@ZIF-8 (Figure 4B,C). Di-peptide-modified ZIF-8 showed a significant improvement of cellular uptake (*p* < 0.001), which was attributable to the high affinity for PSMA on the surface of Lncap cells.

### 3.5. Cell Viability Assay

The cytotoxicity of blank Di-PEG@ZIF-8 was detected by CCK-8 assay. First, Lncap cells were incubated with a series of different concentrations of blank Di-PEG@ZIF-8 for 24 h and 48 h, respectively. Then, the optical density (OD) values were measured with the absorbance at 450 nm using a Synergy H1 Microplate Reader. From Figure 5A,B, it can be revealed that blank Di-PEG@ZIF-8 did not exhibit significant cytotoxicity to Lncap cells either by incubation for 24 h or 48 h, and the cell viabilities were higher than 90%, which indicated that the blank Di-PEG@ZIF-8 was relatively low in toxicity and safe and biocompatible for further use in vivo. Subsequently, the CCK-8 assay was also applied to assess the inhibitory capacity of the various PTX formulations against Lncap cell proliferation after incubation for 24 h or 48 h. As illustrated in Figure 5C,D, the proliferation inhibition of Lncap cells by PTX, mPEG@PTX@ZIF-8, and Di-PEG@PTX@ZIF-8 all exhibited dose-dependent inhibition. Di-PEG@PTX@ZIF-8 could inhibit cell proliferation more effectively at concentrations of PTX ranging from 0.001 µg/mL to 10 µg/mL after 24 or 48 h incubation compared with free PTX and mPEG@PTX@ZIF-8. Furthermore, the IC_50_ value for free PTX, mPEG@PTX@ZIF-8, and Di-PEG@PTX@ZIF-8 were 9.190 µg/mL, 0.1684 µg/mL, and 0.1052 µg/mL after 48 h incubation, respectively. As a result, the encapsulation of PTX by ZIF-8 significantly increased the toxicity to Lncap cells (*p* < 0.01), and the modification with Di-peptide also further increased the toxicity to Lncap cells to some extent. The enhanced antiproliferative effect of Di-PEG@PTX@ZIF-8 was associated with high cellular internalization, as demonstrated in cellular uptake evaluation. Totally, the above results indicated that the Di-PEG@PTX@ZIF-8 displayed stronger cytotoxicity than free PTX and mPEG@PTX@ZIF-8.

### 3.6. Wound-Healing Assay

A wound-healing assay can be performed to simulate tumor cell migration and invasion. As seen in Figure 6, in the control group, Lncap cells proliferated and moved over time, which could lead to a gradual healing of the wounded area. In contrast, the wounded areas in the free PTX, mPEG@PTX@ZIF-8, and Di-PEG@PTX@ZIF-8 NPs groups remained clearly visible and the healing rate was quite low. Table 1 demonstrated the cell migration rate of Lncap cells after treatment with each group and the result suggested that PTX could inhibit the migration and invasive ability of Lncap cells, thus reducing the rate of wound healing. Moreover, the resistance to migration and invasion of Di-PEG@PTX@ZIF-8 was greater than that of free PTX and mPEG@PTX@ZIF-8, which may be attributed to the stronger cytotoxicity of Di-PEG@PTX@ZIF-8 NPs, similar to the above results.

### 3.7. Cell Apoptosis Study In Vitro

Hoechst staining was employed to evaluate the ability of free PTX, mPEG@PTX@ZIF-8, and Di-PEG@PTX@ZIF-8 to induce apoptosis. As illustrated in Figure 7A, normal cells have regular round or oval nuclei, while after treatment with various PTX formulations, the cells appeared to have typical apoptotic-like changes, as shown by chromatin condensation and fragmented nuclei, which displayed dense blue fluorescence after staining. Furthermore, the images demonstrated that the amount of apoptotic-like lesion cells induced by Di-PEG@PTX@ZIF-8 was more than that induced by free PTX and mPEG@PTX@ZIF-8. Subsequently, the Annexin-V/PI double staining method using flow cytometry was employed for further quantitative analysis of cell apoptosis. As shown in Figure 7B,C, the percentages of apoptotic cells in the free PTX, mPEG@PTX@ZIF-8, and Di-PEG@PTX@ZIF-8 group were 33.92 ± 2.334%, 41.17 ± 0.085%, and 56.33 ± 4.594%, respectively. The results established that free PTX, mPEG@PTX@ZIF-8 NPs, and Di-PEG@PTX@ZIF-8 NPs can effectively induce the late apoptosis of Lncap cells, but the apoptotic rates of mPEG@PTX@ZIF-8 NPs and Di-PEG@PTX@ZIF-8 NPs were significantly higher than that of the free PTX group (*p* < 0.01), which was consistent with the tendency of cell proliferation inhibition and also corresponded to the results of cell apoptosis observed under CLSM. Additionally, Di-PEG@PTX@ZIF-8 NPs showed a superior ability to induce apoptosis than mPEG@PTX@ZIF-8 NPs, possibly due to the specific binding of Di-peptide to PSMA receptors on the surface of Lncap cells and increasing the internalization of nanoparticles. All the above results further demonstrated the ability of peptide dimer Di-peptide to target Lncap cells, indicating that Di-PEG@PTX@ZIF-8 has a potent effect on inhibiting Lncap cell proliferation and inducing apoptosis in prostate cancer cells.

## 4. Discussion

With increasing incidence and mortality over the years, more efforts are needed to fight prostate cancer. Traditional antiandrogen therapy, represented by orchiectomy and hormonal drugs, no longer meets the demand for treatment due to its unacceptability and severe side effects. Meanwhile, the rapid development of drug delivery systems has brought new approaches to the treatment of prostate cancer [24]. For example, Yang et al. [25] developed a dual-targeted nanomedicine (PTX-HMNC-EPEG-APSMA) combining specific molecular targeting and external magnetic targeting into a cooperative delivery system for prostate cancer treatment. In addition, due to high porosity, good drug loading capacity, pH responsiveness, and easy modification, ZIF-8 has gradually become a hot spot for drug delivery system research in recent years. The Zn^2+^ contained in ZIF-8 may also prevent the growth of prostate cancer [26] and it has been reported that Zn^2+^ as an adjuvant can enhance the killing effect of PTX on prostate cancer [27]. Furthermore, the specific cancer cell surface antigen PSMA is expressed at 1000-fold higher levels in Lncap cells than in normal cells, and its expression levels increase with disease progression [28]. Therefore, we designed Di-PEG@PTX@ZIF-8 NPs that specifically target the PSMA antigen on the surface of Lncap cells for the treatment of prostate cancer. In terms of the preparation process, Di-PEG@PTX@ZIF-8 NPs can be obtained by a simple one-pot method. Also, the nanoparticles have a suitable particle size distribution and potential, and possess good stability. In vitro release assays have shown that Di-PEG@PTX@ZIF-8 NPs can remain stable in a physiological environment, while rapidly cleaving and releasing the drug in an acidic environment, which means a reduced risk of premature drug leakage and thus fewer toxic side effects. Certainly, the carrier material should have good biocompatibility. The low toxicity of blank ZIF-8 to cells indicates its suitable safety. Due to the modification of peptide dimers, the Di-PEG@PTX@ZIF-8 NPs can specifically recognize the PSMA receptor on the surface of Lncap cells, thus entering the cells by receptor-mediated endocytosis and exerting cytotoxic effects. Overall, the good characterization and cellular-level therapeutic effects shown by the Di-PEG@PTX@ZIF-8 NPs are encouraging for subsequent animal studies. As the disease progresses, primary prostate cancer leads to a decreased sensitivity to androgens in almost all patients and progresses to castration-resistant prostate cancer (CRPC) [29]. It has been shown that the androgen signaling pathway plays an important role in prostate cancer progression even at the CRPC stage, and inhibition of the androgen receptor pathway signaling can improve the antitumor efficacy of chemotherapeutic drugs against CRPC [30]. Also, the delivery of siRNAs via ZIF-8 nanoparticles has been extensively studied [31]. Therefore, we expect that the Di-peptide-modified ZIF-8 system can be applied to the co-delivery of the androgen receptor siRNA and chemotherapeutic agents, which is an option for the treatment of castration-resistant prostate cancer.

## 5. Conclusions

In this study, we constructed novel metal-organic framework nanoparticles loaded with PTX modified with Di-peptide for the targeted treatment of prostate cancer, and evaluated their physicochemical properties and in vitro antitumor effect on Lncap cells. The data showed that Di-PEG@PTX@ZIF-8 NPs possessed appropriate particle size and potential with favorable storage stability. In addition, Di-PEG@PTX@ZIF-8 NPs could remain stable in normal physiological environments, while rapidly cleaving to release drugs by responding to the acidic environment at the tumor site. Also, the carrier material Di-PEG @ZIF-8 did not show any apparent toxicity to cells, and thus displayed a certain degree of safety and biocompatibility. Due to the high affinity of the surface-modified Di-peptide for the PSMA antigen on the surface of Lncap cells, Di-PEG@PTX@ZIF-8 NPs can be highly taken up by the cells and exert significant proliferation-inhibiting and apoptosis-inducing effects. Meanwhile, Di-PEG@PTX@ZIF-8 NPs could effectively inhibit the migration and invasion ability of Lncap cells, thus suppressing the tumor metastasis. On the other hand, since Di-peptide has a specific affinity for prostate cancer cells expressing PSMA such as the Lncap cell line, Di-PEG@PTX@ZIF-8 can exhibit the powerful targeting efficiency for all PSMA-positive cell lines and also significantly increase the targeted therapy outcomes for PSMA-positive prostate cancers. In conclusion, our study suggested that Di-PEG@PTX@ZIF-8 could serve as a novel potential drug delivery system to encapsulate PTX or other chemotherapeutic agents for prostate cancer treatment.

## Figures and Tables

**Figure 1 pharmaceutics-15-01874-f001:**
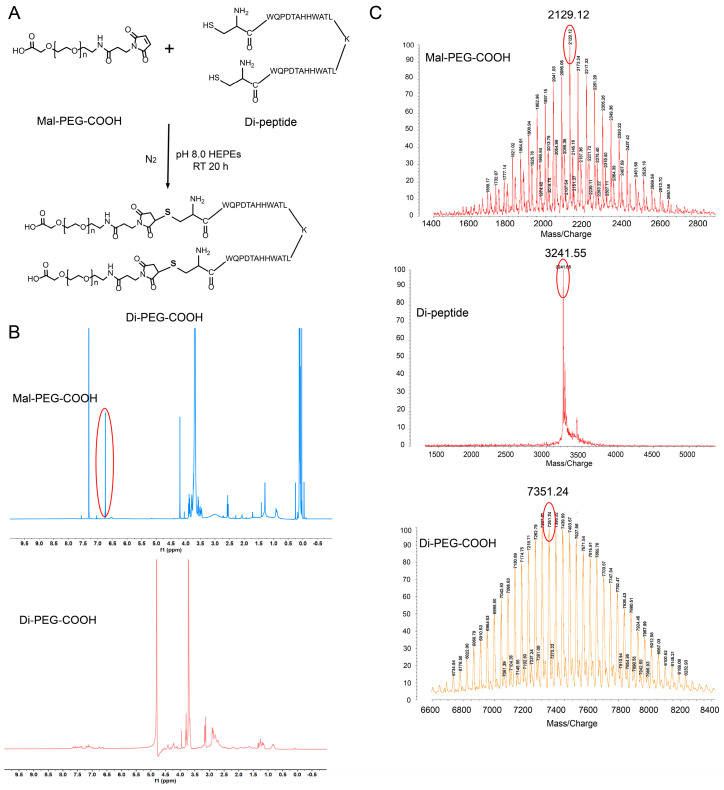
Preparation and characterization of Di-PEG-COOH. (**A**) Synthetic scheme of Di-PEG-COOH. (**B**) ^1^H NMR spectra of Mal-PEG-COOH and Di-PEG-COOH. (**C**) MALDI-TOF-MS spectra of Mal-PEG-COOH, Di-peptide, and Di-PEG-COOH.

**Figure 2 pharmaceutics-15-01874-f002:**
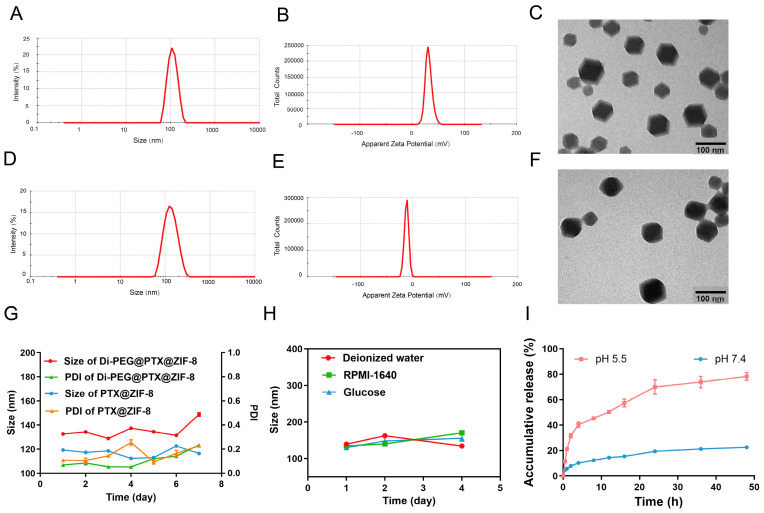
Characterizations of PTX@ZIF-8 and Di-PEG@PTX@ZIF NPs. (**A**) Size distribution of PTX@ZIF-8 NPs. (**B**) Zeta potential distribution of PTX@ZIF-8 NPs. (**C**) TEM image of PTX@ZIF-8 NPs (scale bar: 100 nm). (**D**) Size distribution of Di-PEG@PTX@ZIF NPs. (**E**) Zeta potential distribution of Di-PEG@PTX@ZIF NPs. (**F**) TEM image of Di-PEG@PTX@ZIF NPs (scale bar: 100 nm). (**G**) Storage stability of PTX@ZIF-8 and Di-PEG@PTX@ZIF NPs in PBS at 4 °C. (**H**) Storage stability of Di-PEG@PTX@ZIF NPs in deionized water, RPMI-1640 medium, and 5% glucose. (**I**) The PTX release profiles of Di-PEG@PTX@ZIF NPs in PBS (pH 7.4 and pH 5.5) containing 0.5% Tween-80. The data are presented as the mean ± SD, n = 3.

**Figure 3 pharmaceutics-15-01874-f003:**
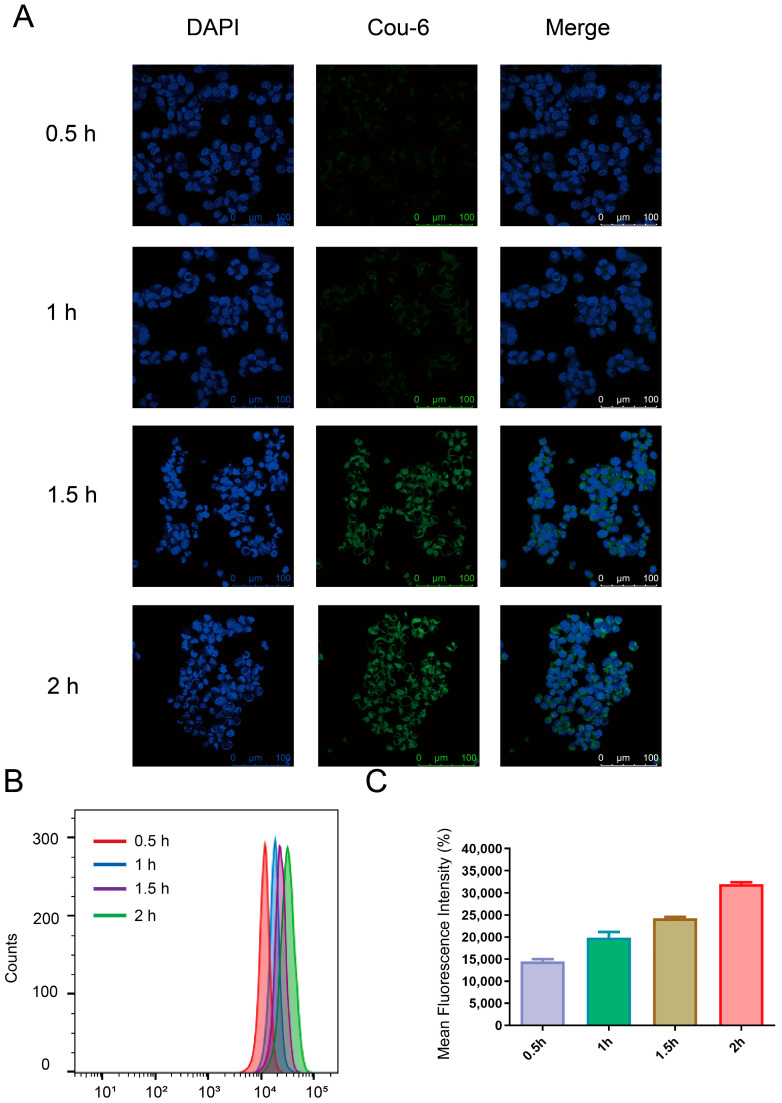
Cellular uptake of Di-PEG@Cou-6@ZIF-8 NPs on Lncap cells. (**A**) Confocal images of cellular uptake after treatment with Di-PEG@Cou-6@ZIF-8 NPs for 0.5, 1, 1.5, and 2 h, respectively. Analysis of cellular uptake of Di-PEG@Cou-6@ZIF-8 NPs for 0.5, 1, 1.5, and 2 h using flow cytometry (**B**) and mean fluorescence intensity (**C**). The data are presented as the mean ± SD, n = 3. Green: coumarin-6, blue: DAPI (nuclei).

**Figure 4 pharmaceutics-15-01874-f004:**
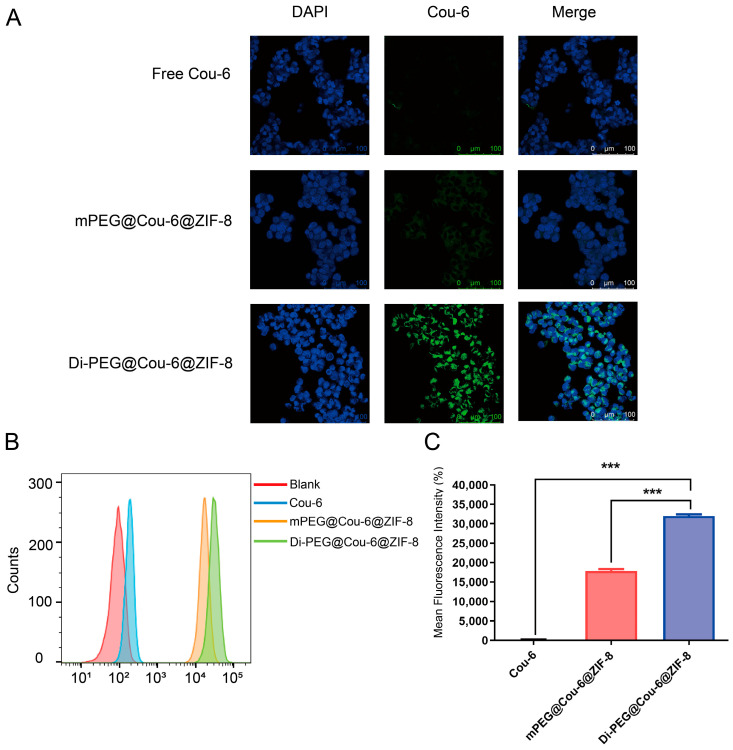
Cellular uptake of free C6, mPEG@Cou-6@ZIF-8 NPs, and Di-PEG@Cou-6@ZIF-8 NPs on Lncap cells for 2 h. (**A**) Confocal images of cellular uptake after treatment with free C6, mPEG@Cou-6@ZIF-8 NPs, and Di-PEG@Cou-6@ZIF-8 NPs for 2 h, respectively. Analysis of cellular uptake of free C6, mPEG@Cou-6@ZIF-8 NPs, and Di-PEG@Cou-6@ZIF-8 NPs for 2 h using flow cytometry (**B**) and mean fluorescence intensity (**C**). The data are presented as the mean ± SD, n = 3; *** *p* < 0.001. Green: coumarin-6, blue: DAPI (nuclei).

**Figure 5 pharmaceutics-15-01874-f005:**
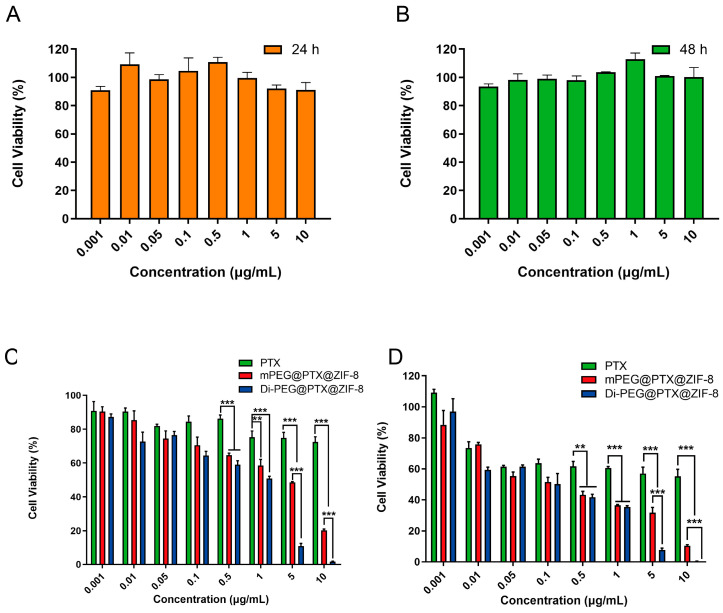
In vitro cytotoxicity of blank Di-PEG@ZIF-8 at an equivalent concentration to Di-PEG@PTX@ZIF-8 NPs ranging from 0.001 μg/mL to 10 μg/mL for 24 h (**A**) or 48 h (**B**) on Lncap cells. The inhibitory capacity of free PTX, mPEG@PTX@ZIF-8, and Di-PEG@PTX@ZIF-8 NPs against Lncap cell proliferation after incubation with different concentrations for 24 h (**C**) or 48 h (**D**). The data are presented as the mean ± SD, n = 6; ** *p* < 0.01, *** *p* < 0.001.

**Figure 6 pharmaceutics-15-01874-f006:**
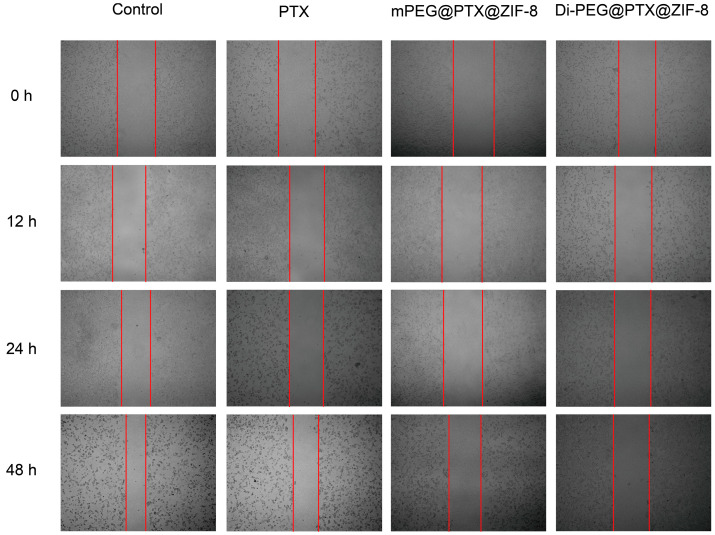
Wound-healing assay of free PTX, mPEG@PTX@ZIF-8, and Di-PEG@PTX@ZIF-8 NPs on Lncap cells.

**Figure 7 pharmaceutics-15-01874-f007:**
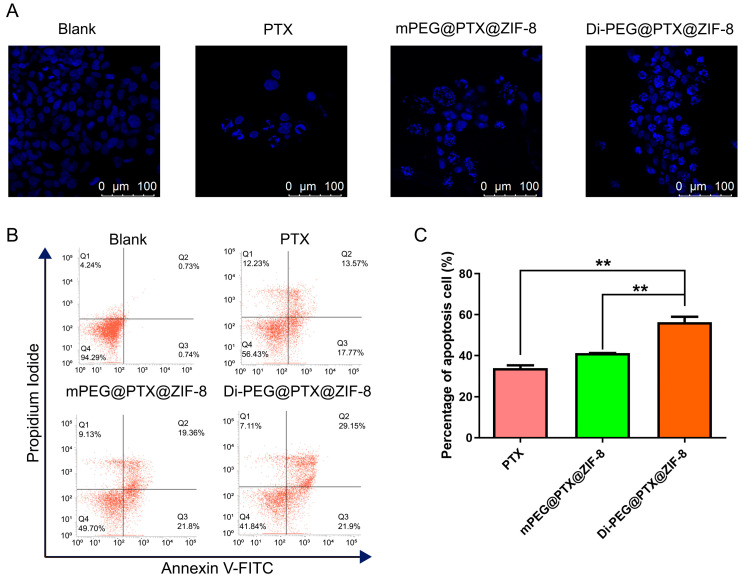
Cell apoptosis on Lncap cells. (**A**) Hoechst staining after 48 h incubation with free PTX, mPEG@PTX@ZIF-8, and Di-PEG@PTX@ZIF-8 NPs, DAPI for nuclei staining (blue), Scale bar = 100 μm. (**B**) The apoptosis assay of Lncap cells analyzed using flow cytometry. (**C**) Quantitative measurements of apoptotic Lncap cells after treatment with different nanodrugs. The data are presented as the mean ± SD, n = 3; ** *p* < 0.01.

**Table 1 pharmaceutics-15-01874-t001:** The cell migration rate of free PTX, mPEG@PTX@ZIF-8, and Di-PEG@PTX@ZIF-8 NPs on Lncap cells.

Time	Control	PTX	mPEG@PTX@ZIF-8	Di-PEG@PTX@ZIF-8
0 h	0	0	0	0
12 h	13.10%	6.00%	1.61%	0.36%
24 h	24.15%	8.19%	4.53%	1.79
48 h	48.45%	31.48	21.63%	3.21%

$Cell\;migration\;rate\;\left(\%\right)=\frac{Scratch\;width_{0\;h}-Scratch\;width_{time}}{Scratch\;width_{0\;h}}\times 100\%$.

## Data Availability

All data available are reported in the article.

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
