# Peer review of "Development of Novel Paclitaxel-Loaded ZIF-8 Metal-Organic Framework Nanoparticles Modified with Peptide Dimers and an Evaluation of Its Inhibitory Effect against Prostate Cancer Cells"

_pharmaceutics, 2023, doi:10.3390/pharmaceutics15071874_

Round 1

Reviewer 1 Report

The article by Zhao et. al titled “Development of novel paclitaxel-loaded ZIF-8 metal organic 2 framework nanoparticles modified with peptide dimers and its 3 inhibitory effect evaluation against prostate cancer cells” investigates the effects of a designer nanoparticle for a potential delivery system targeting prostate cancer. The authors do a nice job explaining this and showing the effects of the nanoparticles on LnCap prostate cancer cells. Some things of note that I noticed when reading their article:

*The authors do not discuss the use of anti-androgens in treatment of prostate cancer. Could this be used in combination with anti-androgen therapy?

*In Figure 2, it is unclear why the authors used the two different pH concentrations that they did? They say one normal pH and one acidic pH of tumor environment. How do they know the specific pH of tumor environment? Is this known in the literature? If so, can they cite studies showing specific pH of tumors.

*The authors only use one cell line. I would ask that they use at least one more prostate cancer cell line besides LnCap cells to determine if it is only specific to one cell type or others that are resistant to anti-androgen therapies.

*Figure 6, wound healing assay is not very convincing. Can the authors measure the differences seen with treatment compared to control and show this in a table or bar graph? Also, was this repeated and reproducible? The Figure legend does not say much?

*Have the authors thought of using a mouse model to test effectiveness?

*It does not appear that the authors included a Discussion section? Is there a reason for this and could they add one that corresponds to similar use of nanoparticles or delivery systems?

Reviewer 2 Report

The manuscript is very interesting and can be accepted after revision 

1.     the overexpression of PSMA on the surface of  Lncap cells was not investigated

2.     Stability of Di-PEG@PTX@ZIF-8 NPs in different media such as DMEM, saline, and distilled water was not studied.

3.     Loading capacity of Di Peptide conjugated NPs was not calculated

4.     In figure 2, one run is not enough to investigate zeta potential surface and diameter. The result should be obtained after 5 successful runs at 25C

5.     cytotoxicity of  Di-475 PEG@PTX@ZIF-8 NPs  was not studied by using healthy cells such as fibroblast or Vero cells and other cancer cells to investigate their target.

6.     Significant values should be written also in the text,  not only in figure caption.

7.     REFs list should have been revised according to pharmaceutics format

English should  revise thoroughly over all the text 

Round 2

Reviewer 1 Report

I think the authors did a good job revising the text, but their new Discussionlacks appropriate references and discussion of similardelivery systems and how theirs would be different and better in prostate cancer.

Additionally, they did not add another cell line, mouse model, etc. or cite appropriate articles such as the Zhang article reference, they gave in their cover letter response.For Conclusions, if this delivery system of the nanoparticle only worksin LnCaP cells then they should specify this in their Conclusions anddiscuss other options for patients with castration resistant prostatecancer.

Reviewer 2 Report

Manuscript revised according to reviewer comments and It is more acceptable now.

A few English typos should have been revised 
